# Plasma-wakefield accelerator simultaneously boosts electron beam energy and brightness

Chaojie Zhang [1] ✉, Douglas Storey [2], Alexander Knetsch [2], Brendan D. O'Shea[2], Robert Ariniello[2], Gevy J. Cao[3], Sébastien Corde [2,4], Thamine N. Dalichaouch[5], Claudio Emma[2], Ole G. Finnerud [3], Spencer Gessner [2], Claire Hansel[6], Elias Hansen [5], Valentina Lee[6], Carl A. Lindstrøm [3], Michael Litos [6], Nathan Majernik[2], Kenneth A. Marsh[1], Warren B. Mori[5], Ivan Rajkovic[2], Mark J. Hogan [2] & Chan Joshi[1] ✉

High-energy particle colliders and X-ray free-electron lasers demand electron beams with qualities currently achieved only in kilometer-scale radio-frequency accelerators. Plasma accelerators promise a compact alternative but have faced challenges in delivering the needed beam quality at relevant energies. Here, we demonstrate that a plasma-wakefield accelerator operating in the nonlinear regime acts as a transformer to simultaneously boost the energy and brightness of an electron bunch injected from the plasma. Using a 10-GeV drive bunch and a three-stage meter-scale plasma source, we generated electron bunches exceeding 20 GeV with sub-percent energy spread, 2 mm·mrad normalized emittance, and multi-kA peak current. A significant number of drive-bunch electrons lost over 90% of their energy, a prerequisite for high energy-conversion efficiency. This demonstration of an energy transformer ratio exceeding two and a brightness enhancement over an order of magnitude opens a path towards cost-effective accelerators for future colliders and light sources.

Advances in conventional radio-frequency (RF) particle accelerators have driven the frontiers of scientific discovery, from probing the subnuclear structure of matter and forces with high-energy colliders[1,2], to enabling femto- and atto-science with ultrabright X-ray free-electron lasers (XFELs)[3,4]. However, these powerful discovery tools require extremely large and expensive facilities, due to the effective electric field breakdown limits (currently at ~100 MV/m) in RF cavities[5]. This limitation has prompted the development of plasma-based accelerators (PBAs)[6], which can support acceleration gradients orders of magnitude larger than conventional approaches[7].

Plasma accelerators use either relativistic charged particle beams (plasma wakefield accelerators, PWFAs[8]) or intense laser pulses (laser wakefield accelerators, LWFAs[9]) to drive large-amplitude plasma waves. In the nonlinear 'blowout' regime[10] of the PWFA, the driver completely expels plasma electrons from its path, forming a nearly spherical ion cavity enclosed by a thin plasma electron sheath moving at nearly the speed of light[11]. This cavity, with radially uniform accelerating fields and linear focusing forces, creates ideal acceleration conditions for electrons[11,12]—essential for preserving beam quality during acceleration.

The plasma acceleration concept has advanced significantly over the past two decades, demonstrating energy gain of ~10 s of GeV in a meter-scale plasma[13–15], a few percent energy spread and high-efficiency energy transfer to a second distinct trailing bunch[16], initial demonstration of energy spread, emittance and charge preservation[17–19], energy reproducibility[20], proof-of-principle staging[21],

[1]Department of Electrical and Computer Engineering, University of California, Los Angeles, CA, USA. [2]SLAC National Accelerator Laboratory, Melno Park, CA, USA. [3]Department of Physics, University of Oslo, Oslo, Norway. [4]Laboratoire d'Optique Appliquée, ENSTA, CNRS, École Polytechnique, Institut Polytechnique de Paris, Palaiseau, France. [5]Department of Physics and Astronomy, University of California, Los Angeles, CA, USA. [6]Center for Integrated Plasma Studies, Department of Physics, University of Colorado, Boulder, CO, USA. ✉e-mail: chaojiez@ucla.edu; cjoshi@ucla.edu

and PWFA as a beam quality transformer[22]. These continuous advancements have recently enabled demonstration of PBA-driven free electron lasing in the extreme ultraviolet (EUV) to optical wavelengths[23–25]. Despite this remarkable progress, plasma accelerators have yet to simultaneously deliver the combination of high energy (multi-GeV), narrow energy spread, and high beam brightness required for applications in hard X-ray FELs[26]. Achieving these beam parameters may also enable a near term demonstration of PWFA technology in a hybrid plasma/RF linear collider[27].

Here we demonstrate that a plasma wakefield accelerator can be used as a transformer to simultaneously boost the energy and brightness of a self-injected electron bunch. Using the 10 GeV electron bunch (Methods), delivered by the recently commissioned FACET-II National User Facility[28] at SLAC National Accelerator Laboratory, and a meter-scale three-stage beam-ionized hydrogen plasma source[29,30], we have generated electron bunches that have more than twice the energy of the drive bunch, a sub-percent energy spread, an order of magnitude smaller emittance, and kA-level peak currents. These parameters yield an energy transformer ratio exceeding two and a brightness enhancement of more than an order of magnitude in comparison to the drive bunch.

## Results

### Experimental setup

Our plasma wakefield transformer consists of a three-stage plasma source (Fig. 1a) that integrates drive bunch focusing, injection of the trailing bunch, and acceleration of the trailing bunch to beyond 20 GeV (Methods). The drive bunch contains variable current spikes (15-100 kA) which induce ionization of hydrogen gas while a longer, lower current base produces the plasma wake[30,31].

The first stage comprises a 1.25-meter-long low-density hydrogen gas region upstream of the gas jet. A 10 GeV electron bunch (1.6 nC charge, 22.6±1.9 mm·mrad emittance, ~20 μm r.m.s. bunch length) from the FACET-II linac operating at 5 Hz was focused at variable positions with respect to the gas jet using quadrupole focusing

magnets with a nominal $\beta^*$ of 50 cm. The high-current spike(s) of the drive bunch[31] partially ionize the hydrogen gas before the drive bunch reaches the gas jet, creating a variable density plasma that acts as a thick, passive lens[32]. This plasma lens further focuses the drive bunch to a few microns (much smaller than its vacuum waist spot size of ~25 μm), increasing the beam density well above the plasma density—essential for exciting a reproducible nonlinear wake[33] even before the drive bunch reaches the gas jet. While this plasma lens section improves the reproducibility of the wake at the gas jet, the position of the of plasma/wake formation with respect to the gas jet varies from shot to shot because of the shot-to-shot variation of the magnitude of the current spike(s) in the drive bunch caused by the amplitude and phase jitter of the RF[30]. As a result, the drive bunch enters the gas jet having lost 1-3 GeV energy to the wake. The second stage is the sharp density downramp with a scale length of a few hundred micrometers, created by a movable blade that intercepts the gas flow at the jet exit (see Fig. 1a), which enables injection of plasma electrons as the phase velocity of the wake's trailing edge suddenly becomes subluminal[34,35]. The third stage consists of the 2.85-meter-long gas region downstream of the gas jet, where the injected electrons are further accelerated until energy of the drive bunch is nearly fully depleted and wake is terminated. The gradual density transition from the higher-density jet to the lower-density gas region self-matches the injected bunch into the acceleration stage[35], as evidenced by the lack of characteristic betatron oscillation features in the bunch's energy spectrum, which will be discussed later. This design eliminates the need for additional plasma-channel guiding[36] or external magnetic elements, as both the unmatched drive and self-matched injected bunches are self-guided by the wakefields throughout the entire meter-scale plasma.

### Drive bunch depletion and 26 GeV injected bunch gain

The energy loss of the drive bunch and energy gain of the injected electrons (Methods) are evident in the energy spectra shown in Fig. 1b–d. These three spectra were measured in different shots while using different spectrometer settings to accommodate the large

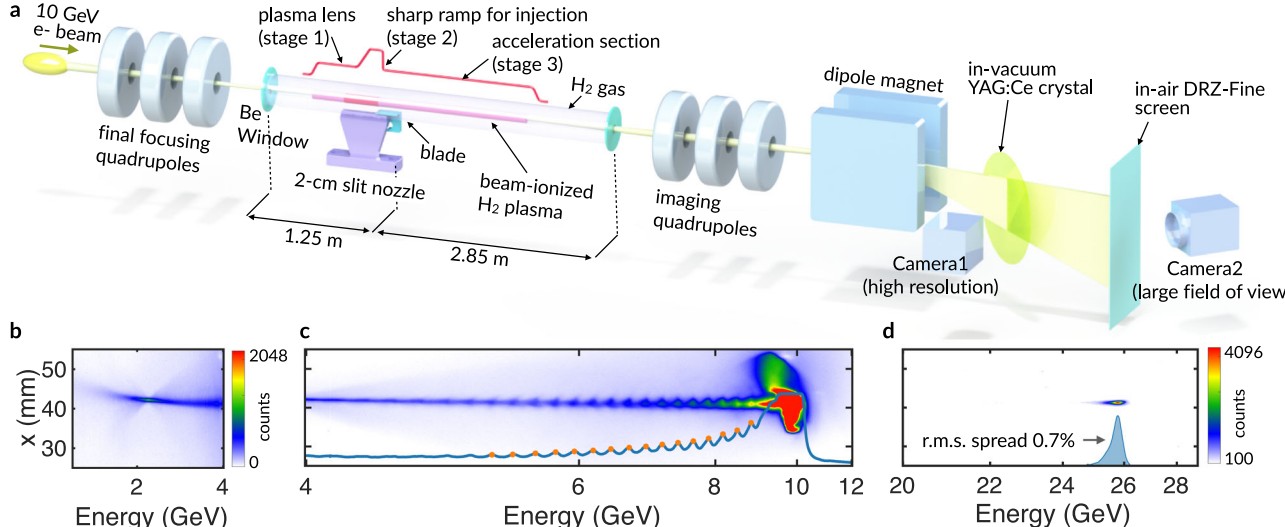

**Fig. 1 | Experimental setup, drive-beam depletion, and self-injection in a meter-scale plasma wakefield. a** A 10 GeV electron drive bunch is focused by quadrupole magnets into a four-meter-long hydrogen-filled region confined by beryllium windows with sub-mm apertures. A two-centimeter-long supersonic gas jet with a movable blade creates a sharp density downramp. As the drive bunch approaches its vacuum waist, its electric field ionizes the hydrogen to form a plasma column (magenta cylinder) and excite a wake. The plasma prior to the gas jet acts as a plasma lens to further focus the drive bunch. Plasma electrons injected at the downramp are accelerated in the wake formed in the subsequent acceleration

section. **b** Energy spectrum showing drive beam depletion to ~1 GeV (at 5.3 Torr, $1.7 \times 10^{17}$ cm$^{-3}$, gas jet off), indicating significant energy transfer to the wake. The spectrometer is focused at 2 GeV. **c** Energy-dependent betatron oscillations of the drive bunch envelope (at 2.4 Torr, $7.8 \times 10^{16}$ cm$^{-3}$, gas jet off) enable determination of the effective plasma length[12]. **d** A distinct injected bunch accelerated to 26 GeV with 0.7% r.m.s. energy spread appears only when the gas jet is activated (at 4 Torr, $1.3 \times 10^{17}$ cm$^{-3}$, gas jet on). For this shot, the spectrometer quadrupoles focus at 15 GeV.

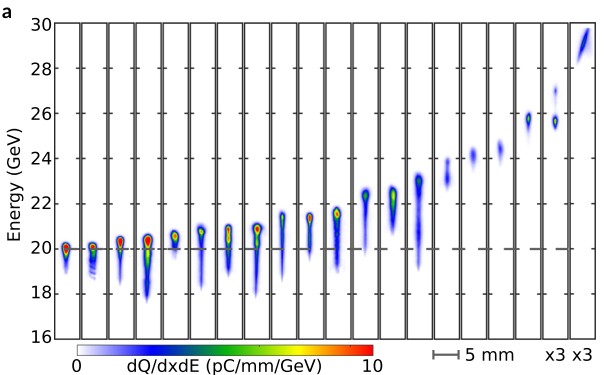

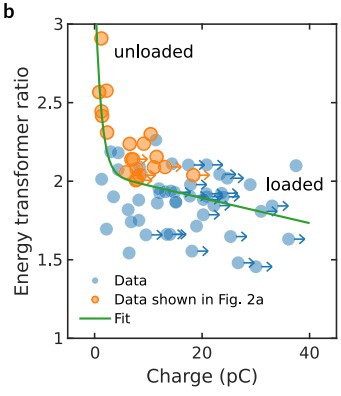

**Fig. 2 | High energy transformer ratio acceleration. a** Twenty shots showing background-subtracted, linearized energy spectra of injected bunches with peak energies exceeding 20 GeV, sorted by energy gain. Data was acquired with the drive beam waist positioned 75 cm downstream of the gas jet and spectrometer focused at 15 GeV. The three highest energy shots were taken with the drive bunch waist at 100 cm downstream. Approximately 15% of shots showed energy gains of >15 GeV in the best datasets. Energy gains of >20 GeV are shown here because these peaks were separated from any acceleration of residual electrons from the tail end of the drive bunch. A maximum energy gain of 29 GeV was seen with first signs of a head to tail tilt of the accelerated bunch. The color bar applies to all shots, with the intensity of the rightmost two shots enlarged by a factor of three for better visibility.

**b** Energy transformer ratio (E-TR) versus injected bunch charge, calculated assuming maximum possible energy loss of 10 GeV by the drive bunch. Orange dots represent shots shown in panel a. Horizontal arrows indicate the actual charge exceeds the plotted value due to camera saturation. The green curve represents a fit combining an exponential decay and a linear term, visually highlighting the transition between different regimes with their distinct slopes. Low-charge shots achieve higher E-TR primarily due to injection closer to the back of the bubble where accelerating fields are stronger, with minimal beam loading preserving these field strengths. The observed data spread results primarily from shot-to-shot variations in the drive bunch current profile, not from the uncertainty in the energy measurement (<1%).

energy difference between the highest energy gain (>20 GeV) and energy loss (≤10 GeV). Figure 1b shows an example of the drive bunch energy loss in the ~1−4 GeV range, where electrons at the peak of the positive electric field of the wake reach the 1 GeV limit of the spectrometer−indicating that some drive bunch electrons have lost 90% of their energy. Charge measurements using beam position monitors (BPMs) immediately downstream of the plasma show that ~200 pC of drive bunch charge was lost, providing additional evidence of energy depletion of the drive bunch, as electrons must lose nearly all their energy before they are overtaken by the wake[37].

Figure 1c displays the energy spectrum in the 4−12 GeV range. While some charge remains at 10 GeV, consisting of electrons ahead of the beam-induced ionization front (previously quantified to be up to 30% of the total charge[30]), the majority of the bunch's interacting electrons experience significant deceleration. The energy loss spectrum exhibits pronounced beam spot size modulations (blue curve), arising from varying divergence angles at different betatron oscillation phases upon exiting the plasma. Applying the analysis method from ref. 12 to these modulations revealed that the 10 GeV electrons undergo ~57 betatron oscillations, while decelerated electrons (9−5 GeV) experience an additional 3-16 oscillations. This corresponds to an effective plasma length of 1.7 ± 0.3 m and a maximum decelerating gradient of 5-7 GeV/m. As we shall see later not all of this plasma length is useful for accelerating the injected particles which are always produced at the downramp.

When the gas jet was activated, a distinct spectral peak appeared (in the case of shot shown in Fig. 1d) at 26 GeV with smaller divergence than the drive bunch. The injected bunch was only observed when the gas jet was turned on and the relative positions of the gas jet and the blade were optimized (to give approximately a factor of two density change in a few hundred μm distance), confirming its origin from the downramp injection process. The bunch in Fig. 1d exhibited an r.m.s. energy spread of only 0.7% and a total charge of 2.3 pC.

## High transformer-ratio acceleration

To change the acceleration length sampled by the injected electrons we positioned the drive bunch vacuum waist at variable distances downstream of the gas jet, effectively shifting the whole plasma, while keeping the injection point fixed at the gas jet downramp. By maximizing the acceleration length (drive bunch vacuum waist 75−100 cm downstream of the gas jet), we achieved electron beam energies exceeding 20 GeV. Figure 2a presents background-subtracted, linearized energy spectra (Methods) of 20 shots, arranged in ascending order of energy, each displaying a well-defined, narrow peak above 20 GeV. Several shots show spectral charge density levels high enough to cause signal saturation on the camera. When we shifted the drive bunch vacuum waist by one meter in four equal 25 cm steps beyond the gas jet, the final energy of the injected bunch increased linearly from ~13 GeV to ~26 GeV. While the wakefield strength may have varied with this change, the linear trend of energy gain with acceleration length indicates that the interaction length was the primary factor determining the final energy, corresponding to an effective average accelerating gradient of 13 GeV/m. Even higher energy but low charge peaks were occasionally seen with a maximum energy up to 29 GeV as shown in Fig. 2a.

We have estimated the energy transformer ratio (E-TR) −defined here as the energy gained by the accelerating bunch divided by energy lost by the drive bunch. The E-TR versus charge contained in the injected bunch plotted in Fig. 2b shows two branches: the first has less than 5 pC of charge but energy higher than 20 GeV, while the second has higher charge (>5 pC) with lower energy. For the low-charge branch, E-TR rapidly increases as the charge decreases due to injection nearer the bubble's high-field region combined with negligible beam loading, whereas for >5 pC charge the E-TR decreases slowly with charge as onset of beam loading occurs. We note that for injected charge >5 pC, approximately half of the shots achieved E-TR exceeding 2, with maximum measured charge of 40 pC. Since linear wakes produced by a longitudinally symmetric drive bunch cannot achieve a transformer ratio greater than two[38,39], our results show that we are in the nonlinear regime. Previous high-gradient nonlinear plasma wakefield experiments with unshaped drive bunches showed E-TRs of only 1-1.5[40,41]. While higher ratios have been demonstrated using shaped drive bunches, those proof-of-principle experiments were limited to ~MeV energy gains using low-gradient acceleration[42,43]. Simulations show that ~3 GeV of the 10 GeV drive bunch energy is lost before it encounters the downramp. Thus, for a bunch accelerated to 20 GeV

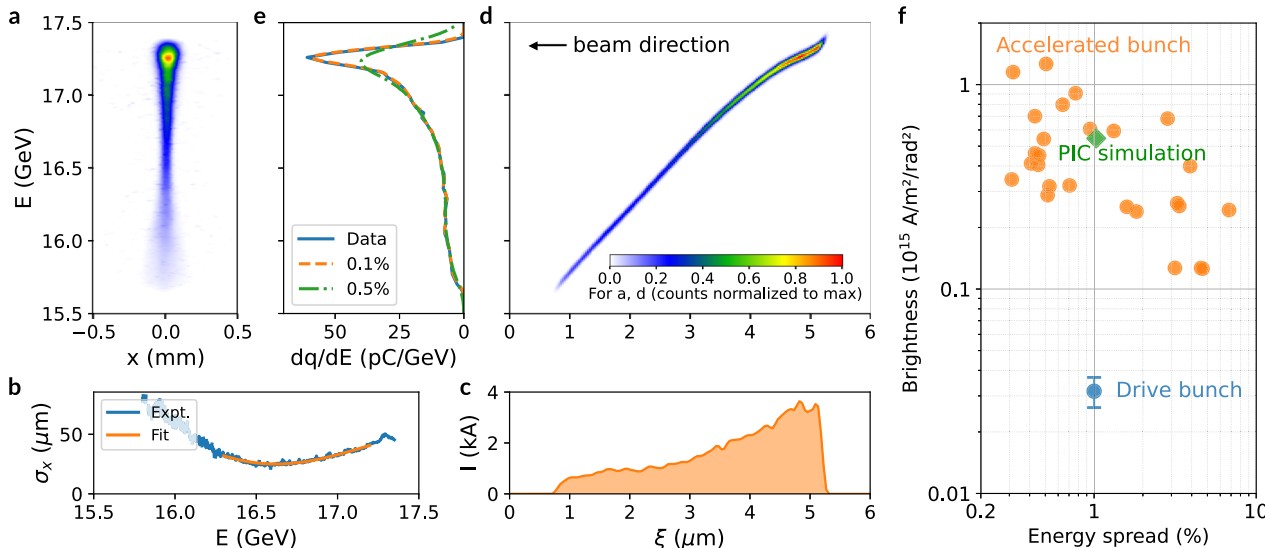

**Fig. 3 | Plasma wakefield accelerator as a brightness transformer. a** Background-subtracted, linearized energy spectrum of a representative shot captured by Camera 1 (see Fig. 1a) with the imaging spectrometer focused at 16.5 GeV. **b** Energy-dependent beam size (blue) and fit (orange), revealing a 2.4 mm·mrad normalized transverse emittance of the injected bunch. **c** Reconstructed current profile of the injected bunch. The bunch has a triangular shape with ~2 μm FWHM length and ~3.5 kA peak current that properly loads the wake to reproduce the measured energy spectrum. **d** Corresponding longitudinal phase space of the injected bunch, showing a nearly linear positive chirp with reduced slope at the back of the bunch.

**e** Energy spectra deduced from the longitudinal phase space with 0.1% (orange dashed) and 0.5% (green dash-dotted) slice energy spread added, showing the former agrees with the measured spectrum (blue) while the latter does not. **f** Brightness versus energy spread for injected bunches (orange) and drive bunch (blue; error bar represents the brightness uncertainty due to measured emittance variations), demonstrating up to 38-fold brightness enhancement. The green diamond represents PIC simulation result confirming the experimental findings (Supplementary Fig. 1).

energy, the E-TR is likely to be up to 3, an estimate that accounts for the ~3 GeV of drive bunch energy lost prior to the injection point. In other words, the E-TR estimates are likely to be an underestimate.

A bunch charge of 10s pC may be adequate for an XFEL, but a future plasma collider will need hundreds of pC charge per bunch for a high drive-to-trailing-bunch energy transfer efficiency and to get the required luminosity at a practical repetition rate[27]. While our current experiments show a maximum injected charge of tens of pC, theoretical beam loading studies using shaped bunches indicate the potential to load several hundred pC with preservation of beam quality[44]. Such a shaped bunch can, in principle, be produced via downramp injection by tailoring the density profile of the ramp[35,45].

## Simultaneous brightness enhancement

We now demonstrate that our plasma wakefield energy transformer also acts as a brightness transformer. Beam brightness is defined as $B_n = \frac{2I_{peak}}{\epsilon_n^2}$, where $I_{peak}$ and $\epsilon_n$ are the peak current and normalized transverse emittance. Clearly, the brightness can be increased by either raising peak current or reducing emittance or both. Here we show that the injected bunch inherently possesses much lower emittance, giving it higher brightness despite lower current than the drive bunch.

Figure 3a presents a high-resolution energy spectrum of an injected bunch captured by Camera 1 (Fig. 1a). The bunch has the smallest waist size at the spectrometer focusing energy of 16.5 GeV in the non-dispersed direction, and the size $\sigma_x$ increases as we move away from this value, as also shown by the blue curve in Fig. 3b. The best fit using the transport matrix of the spectrometer gives a normalized emittance of 2.4 mm·mrad for this shot (Methods). Unlike the drive bunch spectrum (Fig. 1c), the injected bunch shows weak or no beta-tron envelope oscillations, indicating it is matched. Its emittance is an order of magnitude smaller than the drive bunch, and it remains matched to the plasma by gradually decreasing its transverse spot size

$$\sigma_x = \left[\epsilon_n \left(\frac{c}{\omega_p}\right)\left(\frac{2}{\gamma}\right)^{1/2}\right]^{1/2}$$ as it gains energy[46].

To determine the current profile generating the measured energy spectrum, we developed a physics-informed machine learning framework (Methods) to solve the inverse problem. In this framework, we employed a multi-sheath model[47] that accurately describes the plasma wakefield to map an arbitrary injected bunch current profile to a final energy spectrum and iteratively optimized the trial current profile until the calculated spectrum matched experimental measurements (charge is therefore implicitly conserved). The multi-sheath model parameters were adjusted as a function of propagation distance based on PIC simulations to account for wake evolution and dynamic beam loading to improve accuracy. The method was benchmarked using energy spectrum and current profile of downramp injected bunches obtained from PIC simulations, showing good agreement (Methods).

When applied to the experimentally measured energy spectrum (Fig. 3a), this approach revealed the current profile shown in Fig. 3c, which has an approximately trapezoidal shape with ~2 μm (FWHM) bunch length and ~3.5 kA peak current, with a total charge of 25 pC matching the measurement. The corresponding longitudinal phase space (LPS) of the injected bunch is shown in Fig. 3d. The LPS exhibits a nearly linear positive energy chirp with reduced slope at higher energies—a signature of the onset of beam loading. Given that the contained charge is modest and the reconstructed LPS indicates an underloaded wake, the peak current obtained here represents a lower bound of the actual peak current. More than 100 optimizations with different initial current profiles having peak magnitudes ranging from 0.1 to 5 kA converged to similar current profiles as shown in Fig. 3c, indicating that this is a robust solution.

When comparing the energy spectrum derived from this reconstruction (Fig. 3e), assuming a 0.1% slice energy spread reproduces the measured spectrum more accurately than the measured 0.5% r.m.s. projected energy spread, which gives a broader spectral peak with a lower charge density. Using the 0.1% slice energy spread, the longitudinal emittance of the injected bunch is calculated to be ~74 keV·ps, which is comparable to LCLS linac beams (13.6 GeV, ~0.01% slice energy spread[3], 3–16 μm bunch length[48], corresponding to longitudinal

emittances of 14–73 keV·ps). The linear energy chirp can be removed by using either a magnetic chicane to compress the beam to much higher peak current[49], or RF/plasma-based dechirpers[20,50].

The brightness of the injected bunch is calculated using the measured emittance and retrieved peak current, reaching approximately $1.2 \times 10^{15}$ A/m²/rad² for the shot in Fig. 3a. Similar analysis performed for multiple shots is shown in Fig. 3f, with average brightness reaching approximately $0.5 \times 10^{15}$ A/m²/rad². Compared to the beam brightness (blue dot in Fig. 3f) of the drive bunch with 8 kA peak current and $22.6 \pm 1.9$ mm·mrad emittance, the injected bunch is up to 38 times brighter. The experimental brightness values reported here should be considered conservative estimates, as they combine potentially underestimated peak currents with overestimated emittance measurements that likely include growth due to beam transport mismatch outside the plasma. We note that the ~20 GeV peak energy, ~kA peak current, µm-level emittance and $5 \times 10^{14}$ A/m2/rad² (see Fig. 3f and Supplementary Fig. 1) were all reproduced in particle-in-cell simulations (Methods) that model the experiment. Recent simulation studies have shown that leveraging dynamic beam loading effects during pump depletion can accelerate self-injected bunches to above 20 GeV with <1% energy spread and brightness as high as $10^{19}$A/m²/rad², indicating that an improved version of our approach has potential to further boost beam brightness far beyond that of conventional accelerators[51].

## Discussion

We have demonstrated a high-energy plasma wakefield accelerator that simultaneously achieves a sub-percent energy spread and a significant brightness enhancement while pump-depleting the drive bunch. The wake, resulting from charge separation between plasma electrons and ions, functions as a step-up voltage and brightness transformer. By optimally shaping the current profiles of both drive and injected bunches and using a pre-ionized plasma, even higher energy and brightness transformer ratios and increased beam charge should be achievable using a single injector-accelerator stage. This compact, high-energy, high-brightness source thus offers a path toward future coherent light sources and may enable first applications of a plasma wakefield accelerator in advanced collider concepts for high-energy physics.

## Methods

### Drive electron bunch

The upgraded FACET-II National User Facility delivered 10 GeV electron bunches to the interaction point[28]. A photocathode gun generated low-emittance bunches, which were subsequently accelerated through four stages to 10 GeV and compressed using three magnetic chicane compressors installed between acceleration stages. The bunch charge was 1.6 nC with ~2% (r.m.s.) shot-to-shot fluctuation. The normalized emittance was measured to be $22.6 \pm 1.9$ mm·mrad using the quadrupole scan method. Bunch length was measured to be ~20 µm using an X-band Transverse Deflecting Cavity (XTCAV); however, the temporal resolution was insufficient to resolve femtosecond-scale current spikes suggested by beamline simulations[30,31]. These 15-100 kA spikes, containing ≲20% of the total charge, were responsible for ionizing the hydrogen gas but were not the primary driver of the plasma wake. Consequently, charge contained in the current spike (e.g., 0.25 nC) was neglected when calculating the peak current and brightness of the drive bunch (Fig. 3f).

### Plasma source

The plasma was produced by the drive bunch self-ionizing a continuous flow of hydrogen gas. A differential pumping system isolated the 4.1-meter-long gas region from the high-vacuum region of the linac[29]. The gas region was bounded by two 75-µm thick beryllium windows with sub-mm holes that limited gas flow into the high-vacuum

region. These holes were self-drilled by the intense drive bunch and were large enough to allow both drive and injected bunches to propagate through.

Hydrogen gas was delivered using a combination of a pulsed gas jet and an adjustable continuous-flow needle valve to maintain the desired background pressure. The gas jet was created by a custom nozzle that internally transitions from a 3-mm diameter circular input to a slot-shaped throat (2.0 mm by 0.5 mm), and then expands to a 2-cm-long by 2-mm-wide slit exit. The nozzle was positioned ~1.25 m from the upstream beryllium window. For the pump depletion measurements (Fig. 1b, c), the gas jet was turned off and the background pressure was maintained using the adjustable needle valve only. For the injection experiments, the adjustable needle valve was closed, and the pulsed gas jet was operated with a backing pressure of 100 psi at 5 Hz, which maintained the background pressure at 4 Torr, corresponding to a plasma density of $1.3 \times 10^{17}$ cm⁻³ assuming singly ionized hydrogen molecules. A custom-shaped stainless steel blade, with a 10 mm by 4 mm active section and a leading edge machined to 100 µm thickness, covered ~4 mm of the gas jet on the downstream side to create a density downramp. The drive bunch was positioned 2 mm above the nozzle exit. The gas jet operated with 1 ms opening time, carefully timed to maximize injection signal detection probability (~15%). Fluid simulations indicated the downramp had a density ratio of ~2 over a length of several hundred micrometers. The plasma position was controlled by tuning the final focusing quadrupoles to shift the drive bunch vacuum waist (−0.5 to 1.25 m relative to the gas jet), effectively changing the acceleration length since the injection point remained fixed at the downramp location.

### Electron imaging spectrometer

The imaging spectrometer consisted of a quadrupole triplet to capture and refocus electrons exiting the plasma, followed by a dipole magnet to disperse electrons vertically. A high-resolution in-vacuum profile monitor used a cerium-doped yttrium aluminum garnet (Ce:YAG) crystal to convert electron flux into visible light, which was imaged by two cameras with different magnifications: Camera 1 (Fig. 1a) with 3.6 µm effective pixel size and a second camera (not shown) with 30.5 µm pixel size. Additionally, a gadolinium oxysulfide (GOS) scintillator screen viewed by a large field-of-view camera (Camera 2, Fig. 1a) covered the multi-GeV spectral range with 89.1 µm pixel size. These spatial resolutions provided energy resolutions of ~0.01% (Camera 1) and 0.15% (Camera 2). The spectrometer's dispersion curve was calibrated by varying the dipole magnet strength, with an uncertainty of <1% in determining absolute electron energy.

### Injected bunch analysis

Spectrometer images were first background-subtracted using reference images taken without plasma interaction. A region-of-interest analysis isolated the injection signal from the drive bunch contribution using image segmentation. Signal regions were identified by binarizing the image with a threshold of 500 counts for most datasets, with 100 counts used for weaker signals to ensure detection. Among identified regions, the region containing the highest-energy electrons was selected as the injection signal based on energy-weighted pixel intensities. The original pixel intensities within the selected region were preserved for charge calculations. The background-subtracted images were then linearized to account for the nonlinear energy dispersion of the spectrometer (Fig. 2a). The integrated energy spectrum was obtained by summing the signal in the non-dispersive (horizontal) direction.

Charge calibration was performed by comparing the integrated signal from the drive bunch (without plasma) with its known charge measured by upstream beam position monitors. This calibration factor was then applied to the injected bunch signal to determine the injected bunch charge.

The energy spread of the injected bunch was calculated from the linearized energy spectrum using the full width at half maximum (FWHM) divided by 2.355 to convert to an effective r.m.s. spread, which provides a robust measure for non-Gaussian spectral distributions.

## Emittance measurement

The drive bunch emittance was measured using a multi-shot quadrupole scan by varying the spectrometer quadrupole focusing strength to scan the $M_{12}$ element of the transport matrix while keeping the detector plane physically fixed. Electron spot size in the non-dispersive (horizontal) direction was measured by scanning $M_{12}$ from −5 to 5 meters in 11 steps, and the emittance was fitted using[29]: $\sigma_x(E)^2 = \frac{\epsilon_n}{\gamma}\left[M_{11}^2\beta_0 - 2M_{11}M_{12}\alpha_0 + M_{12}^2\left(\frac{1+\alpha_0^2}{\beta_0}\right)\right]$, where all transport matrix elements are functions of beam energy. The drive bunch emittance was measured to be $22.6 \pm 1.9$ mm·mrad without plasma.

For the injected electron bunch, which exhibited larger shot-to-shot jitter, a single-shot emittance measurement was performed using the natural energy spread of the bunch and the highest resolution camera (Camera 1 in Fig. 1a). Since transport matrix elements are energy-dependent, electrons with different energies within the same bunch focus to different spot sizes. By analyzing the correlation between energy and horizontal spot size, the emittance can be extracted from a single shot. Analysis of multiple shots showed injected bunch emittances ranging from 1.9 to 5 mm·mrad with an average of $3.2 \pm 2.0$ mm·mrad.

## Injected bunch LPS reconstruction

We developed a physics-informed machine learning framework to reconstruct the longitudinal phase space of the injected bunch from measured energy spectra. This inverse problem approach allowed inference of critical bunch properties that could not be directly measured with the available experimental setup.

Our method employed a multi-sheath model[47] that accurately described the plasma wakefield, with parameters fitted to match the wakefield structure from our QPAD simulation without injected bunch (see Methods). To account for wake evolution and dynamic beam loading effects, we divided the plasma into 10 longitudinal sections, each with independently fitted multi-sheath parameters, effectively implementing a piecewise-linear approximation of the wake dynamics. This segmentation focused on the first meter of acceleration following injection, where the multi-sheath model remains valid and the injected bunch parameters (energy, energy spread, charge) stabilize (Supplementary Fig. 1c).

The reconstruction followed an iterative optimization process. Beginning with an initial hypothesis for the injected bunch current profile, we calculated the loaded wakefield across all 10 plasma sections using each section's multi-sheath parameters extracted from the QPAD simulation. For each section, we solved the ordinary differential equation governing the bubble shape (with beam loading from the injected bunch), then integrated the resulting accelerating field over that section's length. Energy gains from all 10 sections were summed to obtain the final energy spectrum, which was then compared with the measured spectrum. The current profile was then iteratively adjusted using stochastic gradient descent until convergence between calculated and measured spectra was achieved.

To validate this technique, we benchmarked it against PIC simulations by reconstructing current profiles from simulated energy spectra, finding good agreement between reconstructed and directly simulated profiles (Supplementary Fig. 2). When applied to experimental data, our analysis revealed injected bunches with peak currents ranging from 0.3 to 3.9 kA, scaling roughly linearly with bunch charge.

## PIC simulation

We performed particle-in-cell simulations using two complementary codes: QPAD[52], which employs quasi-static approximation with azimuthal mode decomposition for efficient modeling of meter-scale plasma interactions, and quasi-3D OSIRIS[53] for modeling the down-ramp injection process. Our simulation approach consisted of three sequential stages.

First, using QPAD, we simulated plasma generation through beam ionization and drive bunch dynamics. The simulation employed a moving window of $12 \times 10$ $c/\omega_p$ divided into $1152 \times 960$ cells with a time step of $20$ $\omega_p^{-1}$. Based on previous results[30], the drive bunch was modeled as a superposition of two Gaussian distributions: a current spike (0.25 nC, 0.5 μm bunch length) superimposed onto a base structure (1.35 nC, 20 μm bunch length). The base structure center was positioned 3 $c/\omega_p$ from the window boundary, with the current spike 15 μm ahead. Both components had 10 GeV energy, 20 μm vacuum beam spot sizes, and 50 cm beta functions (corresponding to 16 mm·mrad normalized emittance). A 2.5-meter hydrogen gas region was simulated with the gas jet at 0.6 m and drive bunch focused at 1.0 m. The gas jet region featured a realistic density profile: a 5 mm $\sin^2$-shaped upramp from background density ($1.3 \times 10^{17}$ cm$^{-3}$) to peak density ($2 \times 10^{17}$ cm$^{-3}$), a 1.5 cm uniform region, and a 500 μm $\sin^2$-shaped downramp back to background density. The longitudinal neutral gas density profile and the resulting plasma distribution are shown in Supplementary Fig. 1a and b, respectively.

Second, we extracted the 6D phase space of the drive bunch prior to injection and imported it into an OSIRIS simulation to model downramp injection. We used pre-ionized plasma with the radial distribution extracted from the first QPAD simulation to avoid computationally expensive field convergence calculations and suppress unphysical ionization injection not observed experimentally (no injection signal was observed with gas jet off). This approach improved consistency between the codes' different treatments of field and particle-pushing algorithms. The simulation captured plasma electron trapping as the drive bunch traversed the downramp and initial acceleration over 2.3 mm to tens of MeV energy.

Finally, we imported both injected and drive bunch 6D phase spaces into a second QPAD simulation to model subsequent meter-scale acceleration. The injected bunch charge was adjusted by rescaling charge-per-macroparticle to match experimental measurements while preserving all other beam quality parameters.

Simulations showed that at the exit of the plasma (see Supplementary Fig. 1), the injected bunch was accelerated to ~18 GeV, with 1% projected energy spread (~0.2% slice energy spread), 2.5 mm·mrad normalized transverse emittance, and a ~1.6 kA peak current, giving a brightness of $5 \times 10^{14}$ A/m$^2$/rad$^2$ (green diamond in Fig. 3f).

## Data availability

Source data required to reproduce the figures in this study are freely available in the Zenodo repository[54] under the dataset identifier (https://doi.org/10.5281/zenodo.17260117).

## Code availability

The MATLAB and Python scripts used to generate the figures from the deposited data are available in the Zenodo repository under the dataset identifier (https://doi.org/10.5281/zenodo.17260117). The OSIRIS and QPAD simulation codes are available from the OSIRIS Consortium upon signing a license agreement.

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

## Acknowledgements

The FACET-II E300 and E304 plasma wakefield acceleration experiments were built and operated with funding from the U.S. Department of Energy under Contract No. DE-AC02-76SF00515. This work was supported at UCLA by the U.S. Department of Energy through Grant No. DE-SC0010064. Simulations were performed using resources of the National Energy Research Scientific Computing Center (NERSC), a U.S. Department of Energy Office of Science User Facility located at Lawrence Berkeley National Laboratory, operated under Contract No. DE-AC0205CH11231 using NERSC Award HEP-ERCAP-MP113. S.C. was supported by the ANR (UnRIP project, Grant No. ANR-20-CE30-0030).

## Author contributions

C.Z. and C.J. conceived the experiment. C.Z., D.S., A.K., B.D.O., R.A., S.C., C.E., O.G.F., S.G., C.H., V.L., N.M., and I.R. conducted the experiment. C.Z. and K.A.M. developed the plasma source. C.Z. and C.J. analysed the experimental data. C.Z. and T.N.D. performed the PIC simulations with help from E.H. C.Z. developed the machine learning framework for the beam longitudinal phase space reconstruction. C.Z. and C.J. prepared the figures and wrote the paper, with input from G.J.C., C.A.L., M.L., W.B.M., and M.J.H. All authors discussed the results and contributed to the final manuscript.

## Competing interests

The authors declare no competing interests.
