## [Transparent Peer Review file · Nature Communications]

Plasma-Wakefield Accelerator Simultaneously Boosts Electron Beam Energy and Brightness

Corresponding Author: Professor Chan Joshi

Version 0:

Reviewer comments:

Reviewer #1

(Remarks to the Author)

The authors of this article present and experimentally demonstrate that a plasma wakefield accelerator can simultaneously boost the energy and brightness of an electron beam.

In my opinion, this study is very interesting and shows promising results that can certainly be useful to the scientific community.

Overall, I find the article to be well written and the results to be well presented, with all measurement and analysis methods also well described in the appropriate sections.

Here are some observations, mainly related to the plasma source and its characteristics.

Page 4: When the authors say, "The high current peaks of the propulsion unit partially ionise the hydrogen gas before the propulsion unit reaches the gas jet, creating a variable-density plasma that acts as a thick, passive lens," is there a simulation of this effect available, or a direct or indirect measurement of plasma density due to ionisation peaks? I believe this would certainly add to this deduction.

Page 4: "The position of the onset of plasma/wake formation varies from shot to shot because of the shot-to-shot variation of the longitudinal phase space profile of the drive bunch", is there an estimate of this variation? How was it calculated?

Page 4: "The second stage is the sharp density downramp at the gas jet exit", how long is the second stage? How is the sharp density downramp achieved? With the peak of the driver beam after the modulations? Or because of the blade shown in Figure 1? The latter is not mentioned here in the text.

Page 4: "This design eliminates the need for additional plasma-channel guiding or external magnetic elements as both the unmatched drive and self-matched injected bunches are self-guided throughout the plasma", I believe this depends on the length of the second stage, otherwise we cannot evaluate the effect of self-guiding...

Page 5: "The analysis revealed that 10 GeV electrons undergo approximately 57 betatron oscillations, while decelerated electrons (9-5 GeV) undergo an additional 3-16 oscillations", what analysis are we talking about? It might be appropriate to report it or add a link.

Page 7: "We note that for injected charges >5 pC, about half of the shots reached an E-TR greater than 2, with a maximum measured charge of 40 pC. Since the linear trails produced by a longitudinally symmetric pulse group cannot achieve a transformation ratio greater than two, our results show that we are in a non-linear regime", for the benefit of completeness, I suggest reporting or linking this concept to the application of a model/scaling laws/simulation. The same applies to the end of the same paragraph: "Thus, for a bunch accelerated to 20 GeV energy, the E-TR is likely to be up to 3. In other words, the E-TR estimates are likely to be an underestimate."

Page 11 (section b), plasma source): what are the dimensions and cross-sectional geometry of the section through which the gas is sent via the valve? From the text, we only know its length.

Page 11: "For the injection experiments, the gas jet operated with 100 psi backing pressure at 5 Hz (with the valve closed) to maintain 4 Torr background pressure", does "valve" here refer to the "adjustable needle valve"? Was a fluid dynamics simulation of this condition performed to confirm the result of 100 psi backing pressure?

Page 11: What material and shape does the blade have?

Once revised, I believe this work is suitable for publication, appropriate for the journal, and very useful for the plasma

accelerator community.

Reviewer #2

(Remarks to the Author)

The manuscript by Chaojie Zhang et al. reports experimental results on beam-driven plasma wakefield acceleration. The driver beam has an energy of 10 GeV, while the accelerated electron beam reaches a maximum energy of 26 GeV; its brightness exceeds that of the drive bunch by more than an order of magnitude. Previously, such results could only be achieved using low-gradient acceleration methods. As the authors note, these findings are significant for future x-ray free-electron laser (FEL) sources and particle colliders based on plasma wakefield acceleration (PWFA), where both high acceleration energy and superior beam quality are critical requirements. The current study is highly timely, and the experimental descriptions are generally well-written. While the manuscript merits publication, several conclusions lack clarity and require further refinement.

1. A key factor contributing to the high transformer ratio and brightness of the final accelerated electron beam is downramp injection. I note that the particle-in-cell (PIC) results (see Extended Data Fig. 1) derive from a reduced simulation with a 2.5 m plasma length. However, the downramp injection process is not clearly depicted in this simulation; instead, electrons appear to be injected at the initial stage. If this observation is accurate, the horizontal axis is misaligned and should start from 0.6 m.
2. It would be beneficial to include a clearer visualization of the injection process here. In particular, the gradient of the density downramp in the simulation should be explicitly provided.
3. I am curious as to why the on-axis ion density is lower in Extended Data Fig. 1b.
4. In the first paragraph of Page 4, the authors state: "The gradual density transition from the higher-density jet to the lower-density gas region self-matches the injected bunch into the acceleration stage. This design eliminates the need for additional plasma-channel guiding or external magnetic elements as both the unmatched drive and self-matched injected bunches are self-guided throughout the plasma." However, the claim of self-matching is not clearly substantiated. Significant electron loss may occur in this region, with only the matched electrons undergoing acceleration in the third stage.
5. In the first paragraph of Page 5, the authors assert that "most electrons experience significant deceleration," which is also not clearly demonstrated. Examining Fig. 1c, most electrons appear to remain around 10 GeV (notably, while the gas jet is off and no electrons are injected here, this does not impact the depletion of the drive bunch). The authors should provide a clear energy spectrum of the drive bunch spanning up to 10 GeV, rather than the limited energy range presented in Fig. 1b.
6. The acceleration length is varied by adjusting the drive bunch's vacuum waist, while the downramp injection position remains fixed. As is known, the drive bunch waist affects plasma ionization throughout the entire plasma. This adjustment changes not only the acceleration length but also the injection charge and the wake's acceleration gradient. Attributing the energy tuning of the accelerated electrons solely to the acceleration length is therefore inaccurate.
7. To maintain high beam quality while increasing the final accelerated charge, the authors state that the injected bunch is shaped. However, the underlying mechanism for this shaping is not described.

If the above issues can be reasonably addressed and supplemented, I agree to the publication of the article.

Reviewer #3

(Remarks to the Author)

Version 1:

Reviewer comments:

Reviewer #1

(Remarks to the Author)

The authors present and experimentally demonstrate that a plasma wakefield accelerator can simultaneously enhance both the energy and the brightness of an electron beam. The study is, in my view, of significant interest and provides promising results that are of clear relevance to the scientific community.

The manuscript is generally well written, with the results presented in a clear and structured manner. The experimental methods and data analysis procedures are adequately described in the appropriate sections, allowing the reader to properly assess the robustness of the conclusions. Importantly, the authors have taken into account the reviewers' comments and implemented the requested revisions, which have further strengthened the manuscript and increased its overall value.

In conclusion, I consider this work to be suitable for publication, appropriate for the scope of the journal, and of notable utility for the plasma accelerator community.

Reviewer #2

(Remarks to the Author)

I am very pleased to see that the authors have substantially improved their manuscript in response to the referees' comments. The current results represent an important step forward for PWFA-based accelerators and their potential applications, including FELs. I fully support its publication in Nature Communications.

Reviewer #3

(Remarks to the Author)

Response to Reviewers' Comments

NCOMMS-25-51241A, C. Zhang *et al.*, "Plasma-Wakefield Accelerator Simultaneously Boosts Electron Beam Energy and Brightness".

We thank all the reviewers for their valuable feedback. In the following, we provide a detailed response to their comments.

The reviewers' comments are in black and our responses are in blue.

Reviewer #1

Reviewer Comments: *The authors of this article present and experimentally demonstrate that a plasma wakefield accelerator can simultaneously boost the energy and brightness of an electron beam. In my opinion, this study is very interesting and shows promising results that can certainly be useful to the scientific community.*

Overall, I find the article to be well written and the results to be well presented, with all measurement and analysis methods also well described in the appropriate sections.

Here are some observations, mainly related to the plasma source and its characteristics.

Our response: We thank the reviewer for their positive assessment of the quality of our paper and thoughtful comments on the plasma source. We have addressed each point below.

Reviewer Comment: *Page 4: When the authors say, "The high current peaks of the propulsion unit partially ionise the hydrogen gas before the propulsion unit reaches the gas jet, creating a variable-density plasma that acts as a thick, passive lens," is there a simulation of this effect available, or a direct or indirect measurement of plasma density due to ionisation peaks? I believe this would certainly add to this deduction.*

Our Response: We thank the reviewer for this question. Our statement is supported by PIC simulation shown in the Extended Data Fig. 1b, which shows that high-current spikes in the drive bunch start ionizing the hydrogen gas before the bunch reaches the gas jet. This section of plasma acts as a thick passive lens (length much greater than plasma skin depth), providing additional focusing to the drive bunch.

To provide a clear, quantitative illustration of this effect, we have modified Extended Figure 1b. In the updated figure, we have superimposed the spot size of the drive bunch in plasma (solid blue line) for comparison against its calculated vacuum spot size (dashed black line). This comparison visually confirms the strong focusing effect of the plasma prior to the gas jet, validating its function as a plasma lens.

In addition, the text in the figure caption has been modified as follows:

"The beam size curves clearly show that the best focus position of the electron beam (black-dashed) moves upstream (blue solid) and the beam is more tightly focused by the partially ionized plasma (see color bar of the ion density). Consequently, the gas jet and the subsequent meter-scale hydrogen gas region are fully ionized by the tightly focused electron beam."

Extended Data Fig. 1b is shown below for your convenience:

Reviewer Comment: Page 4: *"The position of the onset of plasma/wake formation varies from shot to shot because of the shot-to-shot variation of the longitudinal phase space profile of the drive bunch", is there an estimate of this variation? How was it calculated?*

Our Response: We thank the reviewer for pointing out this possible confusion. The reviewer is correct that the phrase "variation of the longitudinal phase space profile" was not clear.

The variation we refer to is caused by shot-to-shot amplitude and phase jitter of the RF. This jitter alters the peak current of the fs-scale spikes in the drive bunch exiting the final beam compression stage due to the coherent synchrotron radiation (CSR) effect. A shot with a higher peak current will begin to ionize the hydrogen gas further upstream of the gas jet, while a shot with a lower peak current will begin the ionization closer to the jet. This effect thus causes a variation in the onset position of the plasma. We note that this mechanism and its effects on plasma formation were also observed and documented in our previous work. (see Fig. 2 of Ref. 30).

To clarify this for the reader, we have rephrased the sentence in the manuscript to read:

"The position of the onset of plasma/wake formation with respect to the gas jet varies from shot to shot because of the shot-to-shot variation of the magnitude of the current spike(s) in the drive bunch caused by the amplitude and phase jitter of the RF³⁰."

Reviewer Comment: Page 4: *"The second stage is the sharp density downramp at the gas jet exit", how long is the second stage? How is the sharp density downramp achieved? With the peak of the driver beam after the modulations? Or because of the blade shown in Figure 1? The latter is not mentioned here in the text.*

Our response: The reviewer is correct that while the blade is shown in Figure 1a, its function was not adequately described in the main text where the plasma source stages were introduced.

The sharp density downramp is created by a movable blade that partially intercepts the supersonic gas flow from the 2-cm slit nozzle. This creates a density transition with a scale length of a few hundred micrometers, which constitutes the second stage of the plasma source that enables electron injection.

To ensure this is clear for readers, we have revised the manuscript to explicitly include this information on page 4. The sentence now reads:

"The second stage is a sharp density downramp, with a scale length of a few hundred micrometers, created by a movable blade that intercepts the gas flow at the jet exit (see Fig. 1a)."

Reviewer Comment: Page 4: *"This design eliminates the need for additional plasma-channel guiding or external magnetic elements as both the unmatched drive and self-matched injected bunches are self-guided throughout the plasma", I believe this depends on the length of the second stage, otherwise we cannot evaluate the effect of self-guiding...*

Our Response: We thank the reviewer for this question. There seems to be a misunderstanding, which we are happy to clarify.

The self-guiding mechanism and the "second stage" downramp serve two different functions and occur on different length scales.

- **Self-Guiding:** This effect occurs throughout the entire meter-scale plasma. The high-current spikes of the drive bunch ionize the hydrogen gas, and the resulting wakefield provides a continuous focusing force that guides both the rest of the drive bunch and the injected trailing bunch. This is what eliminates the need for external guiding elements over the full acceleration length.
- **Second Stage (Downramp):** This is a very short region, only a few hundred micrometers long, located at the exit of the 2-cm gas jet. Its role is to inject plasma electrons into the wake.

The self-guiding is therefore not dependent on the length of the second stage. As long as the drive beam has sufficient energy to excite the wake, the focusing field of the plasma ions can focus both the drive and the trailing bunch without the need for external/additional focusing. To prevent this confusion for readers, we have revised the sentence in the manuscript to be more precise:

"This design eliminates the need for additional plasma-channel guiding or external magnetic elements, as both the unmatched drive and self-matched injected bunches are self-guided by the wakefields throughout the entire meter-scale plasma."

Reviewer Comment: Page 5: *"The analysis revealed that 10 GeV electrons undergo approximately 57 betatron oscillations, while decelerated electrons (9-5 GeV) undergo an*

additional 3-16 oscillations", what analysis are we talking about? It might be appropriate to report it or add a link.

Our Response: The analysis, detailed in Ref. [12], deduces the number of betatron oscillations from the energy-dependent modulations of the beam spot size visible on the spectrometer. The principle is that bright spots in the spectrum correspond to electrons exiting the plasma with minimal divergence. The condition that adjacent bright spots are separated by a betatron phase of π allows us to accurately calculate the total number of oscillations.

To make this clearer in the manuscript, we have rephrased this part and now it reads:

"The energy loss spectrum exhibits pronounced beam spot size modulations (blue curve), arising from varying divergence angles at different betatron oscillation phases upon exiting the plasma. Applying the analysis method from Ref. [12] to these modulations revealed that the 10 GeV electrons undergo approximately 57 betatron oscillations, while decelerated electrons (9-5 GeV) experience an additional 3-16 oscillations."

Reviewer Comment: Page 7: "We note that for injected charges >5 pC, about half of the shots reached an E-TR greater than 2, with a maximum measured charge of 40 pC. Since the linear trails produced by a longitudinally symmetric pulse group cannot achieve a transformation ratio greater than two, our results show that we are in a non-linear regime", for the benefit of completeness, I suggest reporting or linking this concept to the application of a model/scaling laws/simulation. The same applies to the end of the same paragraph: "Thus, for a bunch accelerated to 20 GeV energy, the E-TR is likely to be up to 3. In other words, the E-TR estimates are likely to be an underestimate."

Our Response: We thank the reviewer for the suggestion to add a more explicit link to this concept. The principle that a finite length, symmetric drive bunch in a linear wake is limited to a transformer ratio of $TR \leq 2$ is a foundational result in plasma accelerator theory. Our manuscript already cited a relevant discussion in Ref. [38]. However, to improve completeness, we have now also added a citation to the original work that established this theorem: K. L. F. Bane et al., IEEE Trans. Nucl. Sci. 32, 3524 (1985), ref. 39.

Regarding the second point, our statement that the E-TR is an underestimate is based on the ~ 3 GeV energy loss of the drive bunch before the witness bunch is injected. The projection of E-TR of ~ 3 is a direct calculation accounting for this pre-injection energy loss (i.e., 20 GeV / $[10-3]$ GeV). To make this clear, we have revised the sentence in the manuscript to read:

"Thus, for a bunch accelerated to 20 GeV energy, the E-TR is likely to be up to 3, an estimate that accounts for the ~ 3 GeV of drive bunch energy lost prior to the injection point."

Reviewer Comment: Page 11 (section b), plasma source): what are the dimensions and cross-sectional geometry of the section through which the gas is sent via the valve? From the text, we only know its length.

Our Response: We thank the reviewer for this question. The manuscript already specifies the 2 cm by 2 mm slit exit of the nozzle. To provide further detail, the nozzle is fed by a 3-mm diameter circular input orifice. This circular geometry transitions internally to a slot-shaped throat (2.0 mm by 0.5 mm) which then expands through a diverging section to form the final supersonic jet at the slit exit. We have added these details to the Methods section to improve clarity, which reads:

"The gas jet was created by a custom nozzle that internally transitions from a 3-mm diameter circular input to a slot-shaped throat (2.0 mm by 0.5 mm), and then expands to a 2-cm-long by 2-mm-wide slit exit."

Reviewer Comment: Page 11: "For the injection experiments, the gas jet operated with 100 psi backing pressure at 5 Hz (with the valve closed) to maintain 4 Torr background pressure", does "valve" here refer to the "adjustable needle valve"? Was a fluid dynamics simulation of this condition performed to confirm the result of 100 psi backing pressure?

Our Response: The "valve" mentioned refers to the continuous-flow needle valve (which was closed), not the separate solenoid valve for the pulsed gas jet. The 100 psi is the measured backing pressure for the gas jet, not a simulated value.

We have clarified this in the revised manuscript (method section), which now reads:

"For the injection experiments, the adjustable needle valve was closed, and the pulsed gas jet was operated with a backing pressure of 100 psi at 5 Hz, which maintained the background pressure at 4 Torr."

Reviewer Comment: Page 11: What material and shape does the blade have?

Our Comment: The blade is made of stainless steel and has a "Γ" shape to facilitate mounting (see Fig. 1a). The active section that intersects the gas jet is approximately 10 mm by 4 mm with a general thickness of 1 mm. The leading edge that intercepts the gas flow is machined to a minimal thickness of 100 μm. We have revised the Methods section to include these key details, which now reads:

"A custom-shaped stainless steel blade, with a 10 mm x 4 mm active section and a leading edge machined to 100 μm thickness, covered approximately 4 mm of the gas jet on the downstream side to create a density downramp."

Reviewer Comment: Once revised, I believe this work is suitable for publication, appropriate for the journal, and very useful for the plasma accelerator community.

Our Response: We thank the reviewer for their valuable feedback and encouraging assessment. We have addressed all comments point-by-point and hope the revised manuscript is now suitable for publication in Nature Communications.

Reviewer #2:

Reviewer Comment: *The manuscript by Chaojie Zhang et al. reports experimental results on beam-driven plasma wakefield acceleration. The driver beam has an energy of 10 GeV, while the accelerated electron beam reaches a maximum energy of 26 GeV; its brightness exceeds that of the drive bunch by more than an order of magnitude. Previously, such results could only be achieved using low-gradient acceleration methods. As the authors note, these findings are significant for future x-ray free-electron laser (FEL) sources and particle colliders based on plasma wakefield acceleration (PWFA), where both high acceleration energy and superior beam quality are critical requirements. The current study is highly timely, and the experimental descriptions are generally well-written. While the manuscript merits publication, several conclusions lack clarity and require further refinement.*

Our Response: We thank the reviewer for their positive assessment of our work's significance and timeliness. In the following responses, we provide additional details and clarifications to address each of the points raised.

Reviewer Comment: *1. A key factor contributing to the high transformer ratio and brightness of the final accelerated electron beam is downramp injection. I note that the particle-in-cell (PIC) results (see Extended Data Fig. 1) derive from a reduced simulation with a 2.5 m plasma length. However, the downramp injection process is not clearly depicted in this simulation; instead, electrons appear to be injected at the initial stage. If this observation is accurate, the horizontal axis is misaligned and should start from 0.6 m.*

Our Response: We thank the reviewer for identifying that the downramp injection process was not sufficiently illustrated in the original manuscript. This may have led to a misunderstanding about the simulation, which we are happy to clarify.

The results in Extended Data Fig. 1 are from a complete, start-to-end simulation of the entire 2.5-meter beam-ionized plasma, not a "reduced" one. As detailed in the Methods (section g), our simulation is a self-consistent, three-stage process that uses two codes to accurately model the distinct physics of each stage (drive bunch focusing (QPAD), injection (OSIRIS), and acceleration (QPAD)), including the plasma lens effect ($z < 0.6$ m) that precedes injection. The horizontal axis in the figure is therefore correct, as it represents the full interaction distance.

To provide greater clarity and to resolve any ambiguity about the injection location, we have now added a new panel (e) to Extended Data Fig. 1. This new panel, shown below for your convenience, explicitly illustrates the downramp injection process in stage two (the gas jet region). Then the injected electrons are imported into a second QPAD simulation to model the subsequent meter-scale acceleration, which is why in Extended Data Fig. 1c we label the horizontal axis as acceleration length (after injection) instead of the absolute position.

e, Snapshots from the OSIRIS simulation showing the charge density of plasma electrons during the downramp injection process. The frames illustrate the trapping of plasma electrons into a distinct bunch at the back of the wake as the driver traverses the 500- μm -long \sin^2 -shape density downramp, which starts at $z \approx 0.62$ m and drops from 1.5 to 1.0 n_p , where $n_p \approx 1.3 \times 10^{17} \text{ cm}^{-3}$.

We believe this explanation, combined with the new visualization in panel (e), now clarify the simulation method and results.

Reviewer Comment: 2. *It would be beneficial to include a clearer visualization of the injection process here. In particular, the gradient of the density downramp in the simulation should be explicitly provided.*

Our Response: As requested by the reviewer, we have added a clear visualization of the injection process (a new panel (e) in Extended Data Fig. 1). The figure caption has been updated to state that the injection occurs over a 500- μm -long \sin^2 -shaped density downramp that drops from 1.5 to 1.0 n_p , where $n_p \approx 1.3 \times 10^{17} \text{ cm}^{-3}$ is the density of the acceleration stage.

Reviewer Comment: 3. *I am curious as to why the on-axis ion density is lower in Extended Data Fig. 1b.*

Our Response: The on-axis ion density is lower because the simulation models the drive bunch with a Gaussian transverse profile, which has zero transverse electric field on-axis to cause ionization. We have added this to the caption of Extended Data Fig. 1b to clarify.

Reviewer Comment: 4. *In the first paragraph of Page 4, the authors state: “The gradual density transition from the higher-density jet to the lower-density gas region self-matches the injected bunch into the acceleration stage. This design eliminates the need for additional plasma-channel guiding or external magnetic elements as both the unmatched drive and self-matched injected bunches are self-guided throughout the plasma.” However, the claim of self-matching is not clearly substantiated. Significant electron loss may occur in this region, with only the matched electrons undergoing acceleration in the third stage.*

Our Response: We thank the reviewer for this insightful question, which allows us to clarify the evidence supporting our self-matching claim.

Our primary evidence is the lack of betatron oscillations in the injected bunch's dispersed energy spectrum. Unlike the drive bunch (Fig. 1c), which shows strong modulations characteristic of a mismatched beam, the injected bunch's spectrum is smooth (Fig. 2a,

Fig. 3a), indicating that it is matched to the wake. This is further supported by the final emittance, which was measured to be as low as 2 μm . This result is consistent with our simulations (Extended Data Fig. 1c), and the absence of catastrophic emittance growth confirms that the injected bunch is matched.

Regarding the concern about electron loss, our new visualization of the injection process (Extended Data Fig. 1e) shows that once electrons are trapped, they stay phase-locked to the drive bunch and do not get lost during the transition from the injection to the acceleration stage because of the seamless transition.

To make this clearer for the reader, we have added a sentence to the manuscript on Page 4 to clarify this point. It now reads:

"The gradual density transition from the higher-density jet to the lower-density gas region self-matches the injected bunch into the acceleration stage³⁵, as evidenced by the lack of characteristic betatron oscillation features in the bunch's energy spectrum, which will be discussed later."

Reviewer Comment: 5. In the first paragraph of Page 5, the authors assert that "most electrons experience significant deceleration," which is also not clearly demonstrated. Examining Fig. 1c, most electrons appear to remain around 10 GeV (notably, while the gas jet is off and no electrons are injected here, this does not impact the depletion of the drive bunch). The authors should provide a clear energy spectrum of the drive bunch spanning up to 10 GeV, rather than the limited energy range presented in Fig. 1b.

Our Response: The two figures, 1b and 1c, show measurements from separate shots with different spectrometer settings, which were necessary to capture the broad energy range of the decelerated beam. The electrons that remain at 10 GeV are part of the drive bunch but stay ahead of the beam-induced ionization front and thus propagate in neutral gas without interacting with the plasma. As we have shown in previous work, this non-participating charge can be up to 30% of the total [Ref. 30]. Therefore, our statement that "most electrons experience significant deceleration" refers specifically to the interacting portion of the bunch, which is decelerated to energies below 2 GeV. To prevent this confusion for readers, we have revised the relevant sentence in the manuscript to clarify this point. It now reads:

"While some charge remains at 10 GeV, consisting of electrons ahead of the beam-induced ionization front (previously quantified to be up to 30% of the total charge [Ref. 30]), the majority of the bunch's interacting electrons experience significant deceleration."

Reviewer Comment: 6. The acceleration length is varied by adjusting the drive bunch's vacuum waist, while the downramp injection position remains fixed. As is known, the drive bunch waist affects plasma ionization throughout the entire plasma. This adjustment changes not only the acceleration length but also the injection charge and the wake's acceleration gradient. Attributing the energy tuning of the accelerated electrons solely to the acceleration length is therefore inaccurate.

Our Response: We thank the reviewer for this excellent point. The method we used to vary the acceleration length was to tune the final focusing quadrupoles while maintaining the drive bunch's beta function. This is analogous to shifting the focus of a laser by moving an optical lens, which effectively shifts the entire plasma interaction region longitudinally without altering the plasma conditions.

While we agree that small variations in the wakefield could still occur, our analysis shows these were secondary effects. We found that the injected charge remained relatively stable as we varied the waist position, and the measured energy gain exhibits a linear trend with the acceleration length. The plot below summarizes data from several datasets to show the energy gain as a function of various drive bunch waist locations (acceleration length); the blue dots show the energy gain for each individual shot, the orange points show the mean and standard deviation for each waist location, and the green line is a linear fit to the data.

Despite of the shot-to-shot variations, the linear correlation confirms that the change in interaction length was the primary contribution to the energy tunability. We agree with the reviewer that our original wording was probably too strong and have revised the manuscript to improve the clarity. The relevant sentence now reads:

"...When we shifted the drive bunch vacuum waist by one meter in four equal 25 cm steps beyond the gas jet, the final energy of the injected bunch increased linearly from ~13 GeV to ~26 GeV. While the wakefield strength may have varied with this change, the linear trend of energy gain with acceleration length indicates that the interaction length was the primary factor determining the final energy, corresponding to an effective average accelerating gradient of 13 GeV/m."

Reviewer Comment: 7. To maintain high beam quality while increasing the final accelerated charge, the authors state that the injected bunch is shaped. However, the underlying mechanism for this shaping is not described.

Our Response: We thank the reviewer for their question. The reference to "shaped bunches" [43] was intended as a forward-look to future high-charge applications and not a description of our current experiment.

Our downramp injection method does, however, offer a promising path to creating such shaped bunches. As shown in prior work [35], a mapping exists between the initial positions of the plasma electrons in the downramp and their final locations within the

injected bunch once they are trapped. Consequently, the injected bunch's current profile can be controlled by tailoring the downramp's density profile, as demonstrated in our own previous simulation work [Zhang et al., PRAB 22, 111301 (2019), now included as ref. 45]. While we did not actively pursue this in the current experiment, it is a promising direction for future research.

To clarify this for the reader, we have added a sentence to the second last paragraph on page 8, which now reads:

"...While our current experiments show a maximum injected charge of tens of pC, theoretical beam loading studies using shaped bunches indicate the potential to load several hundred pC with preservation of beam quality⁴⁴. Such a shaped bunch can, in principle, be produced via downramp injection by tailoring the density profile of the ramp^{35, 45}."

Reviewer Comment: If the above issues can be reasonably addressed and supplemented, I agree to the publication of the article.

Our Response: We thank the reviewer again for their constructive feedback and positive assessment. We have addressed all of the points raised and hope that the revised manuscript is now suitable for publication.

Reviewer #3:

Our Response: We thank the reviewer for their contribution and appreciate this co-review initiative.